# The Interactive Impacts of Corn Particle Size and Conditioning Temperature on Performance, Carcass Traits, and Intestinal Morphology of Broiler Chickens

**DOI:** 10.3390/ani14050818

**Published:** 2024-03-06

**Authors:** Asadollah Ghasemi-Aghgonbad, Majid Olyayee, Hossein Janmohammadi, Mohammad Reza Abdollahi, Ruhollah Kianfar

**Affiliations:** 1Department of Animal Science, University of Tabriz, Tabriz 5166616471, Iran; majidolyayee@yahoo.com (M.O.); mehrzad.hossein@gmail.com (H.J.); rkianfar@live.com (R.K.); 2Monogastric Research Centre, School of Agriculture and Environment, Massey University, Palmerston North 4410, New Zealand; m.abdollahi@massey.ac.nz

**Keywords:** average daily weight gain, digestive tract, geometric mean diameter, immune responses, villus height

## Abstract

**Simple Summary:**

Published data on the influence of corn particle size (PS) and conditioning temperature (CT) on the performance of broilers are limited. Corn provides a high-energy feed ingredient for broiler chicken diets and is usually ground before being incorporated into mash diets. On the other hand, several factors, including items that are related to assay methodology, bird factors, and dietary factors, e.g., PS and processing, can influence the nutrient utilization of poultry, ultimately affecting the performance of broiler chickens. The aim of this present study was to investigate the interactive impacts of corn PS and CT growth performance, carcass traits, intestinal morphology and immune system in broiler chickens. The results showed that the PS and the CT affected the growth performance of broiler chickens, and this issue shows that feed-producing companies should pay attention to the PS of feed mills, especially corn, as well as the CT.

**Abstract:**

This study aimed to investigate the interactions between corn particle size (PS) and conditioning temperature (CT) on the performance, carcass traits, intestinal morphology, and immune responses in broilers fed a corn-soybean meal-based diet. A total of 360 one-day-old male broiler chicks (Ross 308) were randomly allocated into six dietary treatments in a 2 × 3 factorial arrangement, consisting of two corn PS (finely ground with geometric mean diameter (GMD) of 357 µm (PS_F_) vs. coarsely ground corn with GMD of 737 µm (PS_C_), and three CT [unconditioned (CT_U_), conditioned at 75 °C (CT_75_) and 90 °C (CT_90_)]. Birds were accommodated in 30 pens with five replicates and 12 chicks per each pen. There was no interaction between corn PS and CT on the growth performance and immune response of broilers at any growth phases. However, during the starter (0–10 days) period, the average daily weight gain (ADWG) and feed conversion ratio (FCR) of PS_F_-fed birds were significantly improved compared to those fed PS_C_ (*p* < 0.05). During the starter (0–10 days) and grower (11–24 days) periods, increasing the conditioning temperature of corn increased the ADWG, while in the starter phase only the CT_75_ caused a lower FCR (*p* < 0.05). Broilers fed PS_F_ corn showed the lowest FCR during the finisher (25–42 days) period compared to those fed PS_C_ (*p* < 0.05). Conditioning corn at 75 °C reduced FCR during the finisher (25–42 days) period compared to the birds fed CT_U_ and CT_90_ corn (*p* < 0.05). In whole experimental periods (1–42 days), PS_F_ and CT_75_ treatment increased the ADWG compared to the PS_C_ and CT_U_ (*p* < 0.05). The CT_75_ treatment improved primary total anti-sheep red blood cell (SRBCs) titer (IgT) and IgM and secondary IgT and IgG responses compared to the other experimental groups (CT_U_ and CT_90_) (*p* < 0.05). No significant PS × CT interaction was found on the Newcastle disease (ND) antibody titer of broiler chickens (*p* > 0.05). Feeding CT_75_ corn reduced duodenum and jejunum relative lengths compared to the birds fed diets containing CT_U_ corn. Significant PS × CT interactions (*p* < 0.05) were observed for villus height, villus height to crypt depth, crypt depth, muscle thickness, and absorption surface area of the jejunum. The highest carcass yield was observed in the PS_F_-CT_75_ group (*p* < 0.05). In conclusion, the use of finely ground corn (PSF) conditioned at 75 °C (CT75) was beneficial to growth performance, development of the digestive tract, jejunum histomorphometry and the immune responses of broilers.

## 1. Introduction

Corn (*Zea mays* L.), the primary dietary energy source in monogastric animals, especially poultry, is the dominant cereal grain produced worldwide, and its non-processed usage in the manufacture of bio-based products is increasing. Due to its better nutritional value, especially the excellent source of carbohydrates with low fiber and soluble non-starch polysaccharide content, corn grain has gained many attractions (compared to wheat and barley) in poultry nutrition [1].

Due to several advantages, such as improving nutrient digestibility, energy utilization and increasing growth performance and production economics, feed processing has become an essential step in the poultry feeding system [2]. Processing quality, nutritional value of the feed ingredients, performance, and gastrointestinal tract (GIT) function have been shown to be directly affected by grain grinding degree [3]. According to recent studies, FCR and the utilization efficiency of nutrients improved in broiler chickens fed diets containing coarsely ground corn grain [4]. Finely ground cereals have been shown to increase nutrient digestibility because of their high relative surface area, which interacts better with digestive enzymes in the GIT [5,6]. This improvement in digestibility may be attributed to better digestion (enzymatic) and absorption processes in the small intestine [6,7].

Today, different feed-processing techniques have been introduced to improve nutrient digestibility in the poultry industry. For instance, physical and chemical changes caused by steam conditioning have been reported to affect nutrient utilization positively [8]. Earlier studies have shown that the moderate steam conditioning of broiler feed can positively affect its nutritional value through (a) starch gelatinization, (b) heat-sensitive anti-nutrient destruction, (c) cell wall destruction, (d) digestive enzyme inhibitor protein denaturation, and (e) nutrient availability improvement. In contrast to moderate conditioning, high processing temperatures may impair the nutrient digestibility of broiler feed by, for example, triggering the Millard reaction [8].

The hypothesis of this study was that feeding birds with different corn particle sizes and conditioning temperatures may improve their performance potential and enhance their gastrointestinal development. The present experiment evaluated whether fine or coarse ground corn and different conditioning temperatures would influence growth performance, carcass traits, intestinal morphology and the immune system in broiler chickens.

## 2. Materials and Methods

### 2.1. Experimental Design and Processing

The authors confirm that the ethical policies of the journal, as noted on the journal’s author guidelines page, have been adhered to and the appropriate ethical review committee approval has been received (in line with the Animal Research Ethics Committees of the University of Tabriz under the ethical approval ID of IR.TABRIZU.REC.1402.130 and approval date: 7 January 2024. The authors confirm that they have followed EU standards for the protection of animals used for scientific purposes (and feed legislation, if appropriate). This experiment was conducted to investigate the effects of corn processing in broiler diets at different CT and PS on their growth performance, carcass characteristics, immune responses, and small intestine morphology in a completely randomized design with a 2 × 3 factorial arrangement including two corn PS: fine corn PS (PS_F_) (357 ± 15 µm) and coarse corn PS (PS_C_) (737 ± 19 µm) and three different CT of corn (unconditioned (CT_U_), 75 °C (CT_75_) and 90 °C (CT_90_)). The corn grain used in the current study was obtained from the nearest local feed supplier in West Azerbaijan, Iran. In this experiment, the chemical composition of corn was first determined [9]. Then, the fine and coarse particle sizes of corn were obtained using a hammer mill with two and eight mm hole screens, respectively. An eight-sieve stack with the sieve numbers of 4, 6, 10, 18, 35, 60, 120, 200, and a pan was used to determine corn grain particle size. Approximately 100 g of the ground corn samples were sifted for 10 min. The particles retained in each sieve were weighed and expressed as the proportion of the initial sample weight. In line with the methods of Baker and Herman [10], the geometric mean diameter (GMD) and the geometric standard deviation of corn particle size were then determined. After preparing the corn with a specific PS, each of the fine and coarse particles was divided into three equal aliquots. The first part remained unprocessed; the other two parts were conditioned separately at 75 and 90 °C for 60 s in a commercial double conditioner (Eghtesadgostare Salem Poultry Feed Co., Urmia, Iran) with an adjustable steam flow rate. A two-bar steam pressure was used for conditioning. It should be mentioned that, in order to have a better and more uniform distribution, the steam entered through six-steam injection valves at the beginning of the conditioner. Following the heat treatment to the ground, the corn was dried and cooled at 25 °C for eight min in the cooler. The six differently processed corn batches were used to develop dietary treatments in mash form.

### 2.2. Diets and Bird Management

A total of 360 one-day-old, male Ross 308 broilers were purchased from the nearest commercial hatchery and settled at the experimental farm from one to 42 days of age. The birds were allocated into six treatments with five replicates and 12 chicks per each, in floor pens with the dimensions of 120 × 120 cm^2^. The experimental diets were fed in three phases, namely the starter (1–10 days), grower (11–24 days), and finisher (25–42 days) phases. The feed was provided in mash form and the birds had free access to fresh water throughout the study. The ingredients and nutrient compositions of the experimental diets are shown in Table 1.

The stocking density of 12.5 birds/m^2^ was considered during the study. The brooding temperature was maintained at 32 °C for the first day and was gradually decreased by 1 °C every three days until 21 °C and maintained to the end of study. The lighting program, relative humidity, and temperature followed the Ross 308 management guide [11], and the diets were formulated to meet the nutrient recommendations for Ross 308 [12].

### 2.3. Sample and Data Collection

#### 2.3.1. Growth Performance

Growth performance traits, including ADWG (Average Daily Weight Gain) and ADFI (Average Daily Feed Intake), were measured at the end of each feeding phase, and the FCR was calculated by dividing the ADFI by ADWG using the following formula:ADFI(g/bird/d) = cumulative feed intake (g)/(number of birds × number of days)(1)
ADWG(g/bird/d) = final weight gain − initial weight/number of days(2)
FCR (g/g) = ADFI(g)/ADWG (g)(3)

Mortality was recorded daily. FCR values were corrected for the body weight of any bird that died during the course of the experiment.

#### 2.3.2. Carcass Traits

On d 42, 10 chickens with similar BW from each treatment (two bird per each replicate) were selected, weighed, and euthanized by cervical dislocation. Then, breast muscle, thigh, wing, heart, liver, gizzard, proventriculus, pancreas, small intestine segments, abdominal pad fat, thymus, spleen, and bursa of Fabricius were taken out and weighed. After digesta depletion, the length and weight of small intestine segments including duodenum, jejunum and ileum, and cecum were measured. The relative empty weight of organs and length of the intestine segments were calculated as follows:Relative empty weight of organs = (Empty weight of organ (g)/slaughter body weight (g)) × 100(4)
Relative intestinal length (cm/kg body weight) = Intestinal length (cm)/body weight (gr)(5)

#### 2.3.3. Jejunum Morphology

At d 42, two birds per each replicate whose body weight was closest to the mean weight of the pen, were selected, weighed and euthanized by cervical dislocation. Briefly, 1.5 cm from the middle of the jejunal segment was isolated, and following the tissue fixation with 10% formaldehyde, they were dehydrated with graded ethanol and embedded in paraffin wax. Four cross-section cuts with 4 µm thickness were provided and stained with hematoxylin and eosin (H&E). The histomorphometry indices of jejunum (villus height, villus width, crypt depth, villi-to-crypt ratio, crypt depth ratio, and absorption surface area) were analyzed using a light microscope (Olympus Model BX41, Japan) and Digimizer image analysis software (V.6.3.0). The villus absorption surface area was calculated using the formula [13,14]:Villus absorption surface area = 2π × (average villus width/2) × villus height(6)

To evaluate the muscle layer thickness of the broiler intestine, the circular and longitudinal muscular layers thickness was measured at 10-fold objective by using a light microscope (Olympus BX 41, Japan).

#### 2.3.4. Immune Responses

Sheep red blood cell (SRBC), a non-specific and T-cell-dependent antigen, was administered to quantify antibody production. Briefly, for primary antibody titers to SRBC, 2 male broiler chickens per treatment were randomly selected at 28 d of age, and 1 mL of a 5% SRBC suspension diluted in phosphate-buffered saline (PBS) was intramuscularly injected into breast muscle. At 35 d of age, a booster inoculation of 1 mL of 5% SRBC was conducted on the same birds. For the secondary humoral immune response, the same SRBC solution was injected intramuscularly into the breast muscle of the same birds at 35 d of age. Blood samples were collected 7 days after each injection, and serum was collected to determine total IgM and IgG anti-SRBC antibody titers by microhemagglutination assay [15].

At 42 d of age, serum antibody titers against Newcastle disease (ND) virus were performed using a standard hemagglutination inhibition test [16].

### 2.4. Statistical Analysis

The data obtained in the present study were analyzed using the GLM procedure of SAS 9.2 software [17] in a two-way factorial arrangement following model:Yij = µ + Ai + Bj + (AB)ij + εij(7)
where Yij was the observed dependent variable; µ = the overall mean; Ai = the effect of PS treatment; Bj = the main effect of CT treatment; (AB)ij = the interaction between PS and CT; εij = the random error.

Pens were considered as the experimental unit for all data. The significance of the differences among means was determined using Duncan’s Multiple Range Test, where the *p* < 0.05 was considered statistically significant.

## 3. Results

### 3.1. Growth Performance

The effect of different corn PS and CT on the growth performance of broiler chickens is reported in Table 2. The results showed no significant (*p* > 0.05) interaction between PS and CT on the growth performance of broiler chickens at any growth phases. However, PS_F_ significantly (*p* < 0.05) improved FCR and ADWG in the starter (1–10 days) and whole experimental period (1–42 days) compared to birds receiving PS_C_ diets. During the finisher phase, the FCR in PS_F_-fed birds was significantly lower (*p* < 0.05) compared to PS_C_-fed birds. The main effect of CT was significant (*p* < 0.05) for ADWG and FCR in the starter phase. In the grower phase, the main effects of CT were significant in ADWG and FCR, where the highest ADWG and lowest FCR were observed in CT_75_ and CT_90_ (*p* < 0.05). In the finisher (25 to 42 days) phase, the main effect of different CTs was significant on FCR, with the lowest FCR observed in CT_75_ (*p* < 0.05). In the whole grow-out period, the main effect of CT was significant (*p* < 0.05) for ADWG, with CT_75_ having the highest ADWG.

### 3.2. Relative Length of the Small Intestine and Jejunum Morphology

The effects of different corn PS and CT treatments on the relative length of the small intestine and jejunum morphology of broiler chickens are presented in Table 3. As shown, the PS × CT interaction was significant for the cecum relative length, villus height, crypt depth, villus height to crypt depth ratio, muscle thickness, and absorption surface area. The highest relative length of cecum and crypt depth was observed in birds fed the PS_F_-CT_U_ diet (*p* < 0.05). The birds with the highest villus height and absorption surface area belonged to those fed the PS_F_-CT_75_ diet (*p* < 0.05). The thickest intestinal muscle was observed in PS_C_-CT_U_-fed birds (*p* < 0.05). The main effect of PS was significant on GIT, where the highest villus height, villus height:crypt depth, and absorption surface area were observed in PS_F_ (*p* < 0.05). The highest villus width, crypt depth, and intestinal muscle thickness belonged to the PS_C_ (*p* < 0.05). The main effects of different CT treatments on dietary corn grain were significant in GIT, where the lowest percentages of duodenum, jejunum, cecum, total small intestine and muscle thickness (μm) were observed in CT_75_ (*p* < 0.05). In the meantime, the highest villus width and absorption surface area belonged to CT_75_ and CT_90_ (*p* < 0.05) compared with those fed CT_U_.

### 3.3. Immune Responses

The effects of experimental treatments on the broiler chickens’ responses to SRBC and ND titers are shown in Table 4. The interaction between PS and CT was not significant on SRBC and ND titers (*p* > 0.05). The main effects of CT_75_ on the SRBC titers of broilers were significant, with the highest IgM and IgT titers in the primary response and IgG and IgT titers in the secondary response against SRBC injection observed in CT_75_ (*p* < 0.05).

### 3.4. Carcass Traits

The effects of corn PS and CT on the carcass characteristics of broiler chickens are presented in Table 5. As indicated, significant PS × CT interaction was seen in carcass yield, gizzard, liver, and pancreas relative weight (g per kg body weight), where the highest carcass yield was observed in birds fed PS_F_ corn conditioned at 75 °C (*p* < 0.05). Moreover, the highest liver relative weight belonged to the PS_F_-CT_U_-fed birds, and the highest pancreas weight belonged to those fed the PS_F_-CT_U_ and PS_F_-CT_75_ diets (*p* < 0.05). The main effect of PS on dietary corn grain on body weight was significant, with the highest value observed in PS_F_ (*p* < 0.05). The main effect of PS on the abdominal fat pad, gizzard, pancreas, and gizzard pH was significant, with the highest value seen in the PS_C_, PS_C_, PS_F_ and PS_F_ groups, respectively (*p* < 0.05). The main effects of CT_75_ and CT_90_ treatments on gizzard and pancreas relative weight were significantly higher (*p* < 0.05) than in the CT_U_ group.

## 4. Discussion

### 4.1. Growth Performance

According to the findings of the current study, the main effect of the corn PS on the growth performance of broiler chickens was significant, where the PSF treatment improved BWG in comparison to the birds’ diets containing PSC-treated corn grain. Studies have shown that feeding finely ground diets can increase the growth performance of broiler chickens. Reducing the particle size of grains used in animal feeds means fracturing the outer seed coat and its endosperm layer. Finely grinding grains increases the particle numbers while reducing their size. Therefore, the overall surface area per unit of volume increases, which in turn increases the accessibility of digestive enzymes and consequently increases digestive efficiency [18]. Also, easy handling and better mixing with the other feed ingredients are the other advantages of finely ground PS [19]. Grain particle size is crucial in feed processing. It has been shown that bigger particles alleviated the intestinal digesta passage rate, which in turn increased peristaltic movements and improved nutrient utilization [20]. On the other hand, studies showed that grinding broiler chickens’ feeds influenced their nutrient digestion by decreasing surface area, which subsequently may enhance feed intake and gut health [21]. There are reports that broiler chickens preferred coarse particles-containing diets to those of fine particles [22]. However, it is still not clear whether coarse-sized feeds are beneficial for all ages of birds. The results obtained in the present study clearly demonstrated that finely ground corn grain inclusion in broiler diets improved ADWG and FCR during the starter phase (1–10 days); however, it only improved FCR in the finisher period (26–42 days) and increased ADWG in the whole experimental period (1–42 days). No significant impact was observed in the grower period (11–25 days) on the birds’ performance. These findings are in line with those obtained by Chewning et al. [23], who reported no remarkable difference in BWG phases by feeding diets containing coarsely-ground corn grain (600 μm) and finely-ground corn grain (300 μm). Similar results were reported by Get et al. [21], where they showed that finely ground corn particle size in the starter and growth phases and course one in the finisher phase were beneficial for BWG in broiler chickens. These researchers suggested that grinding the corn grain finely increases the contact surface of the feed with digestive enzymes and ultimately improves the ADWG and FCR.

The conditioning temperature significantly affected the FCR and ADWG in the starter phase, where the lowest FCR and the highest ADWG in the starter period were observed in CT_75_. Briefly, CT_75_ treatment of corn grain in this study improved the FCR and ADWG in all experimental phases except for the grower (11–24 days). Abdollahi et al. [24] have reported improved BWG as conditioning temperatures increased from 60 to 75 and 90 °C in broiler chickens. Also, according to the study of Selle et al. [25], increasing the conditioning temperature from 65 to 95 °C in sorghum-based diets increased the broiler chickens’ FCR linearly with no adverse effect on feed consumption. Recently, Netto et al. [26] reported a quadratic effect of CT on BWG, where conditioning feeds at 60 and 70 °C improved the broilers BWG at 21 d of age in comparison to those fed diets conditioned at 90 °C. Those researchers declared that the conditioning temperature elevation may reduce the heat-labile nutrients digestibility [27,28,29] by playing a role in the indigestible starch-protein and starch-lipid complex formation, which may deteriorate nutrient absorption in the small intestine [26,30].

### 4.2. Carcass Traits

In the present study, the two factors of corn PS and CT had no significant effect on carcass traits, including carcass weight, breast weight, and thigh weight. Only the finely ground corn led to a significant reduction in abdominal fat pad compared to diets containing coarsely ground corn grain. Yan et al. [31], in agreement with the findings of the current study, reported that abdominal fat increased as the feed particle size increased. Similarly, Rezaeipour and Gazani [32] found that different feed particle sizes did not affect the breast and thigh relative weights. Also, Massuquetto et al. [33] reported that carcass yield, breast, thigh, and drumstick yields were not affected by the physical form of diet. Interestingly, the present work showed that finely ground corn grain inclusion in the broiler diet significantly decreased the abdominal fat pad in comparison to those fed diets containing coarsely ground corn. These results are in contrast to those obtained by Mingbin et al. [34], where the authors state that the physical nature of the diet has significant effects on broiler carcass yield. However, Unni et al. [35] found that the slaughter yield of broiler chickens may be affected by different feed particle sizes.

According to the present findings, different CTs of corn grain had no significant impact on the carcass characteristics of broiler chickens. However, Rueda et al. [36] reported that conditioning broiler chickens’ diet at 82 °C increased their body weight compared to those fed diets conditioned at 71 to 77 °C; however, the differences were not statistically compared to those conditioned at 88 °C. According to the reports of Loar et al. [29], carcass traits of broiler chickens were not affected by the conditioning temperatures of corn grains (74, 85, and 96 °C) or the amount of fat added to the mixer (1.00 and 2.18%). Similarly, Cutlip et al. [37] observed that the 39 day-old broiler chickens fed diets conditioned at different temperatures (82.2 and 93.3 °C) and steam pressures (20 and 80 psi) had no significant differences in their breast yield and fat pad.

### 4.3. Immune Responses

As shown in Table 4, the immune indices in primary and secondary responses against SRBC were increased by conditioning at 75 °C at 42 d of age. Currently, broiler chickens are genetically selected for improved FCR and rapid growth rate. Increased body weight gain has been reported to be negatively correlated with antibody response (total antibody response and SRBC) in broiler chickens. As recently discussed in detail by Kiarie and Mills [38], it has been well proven that the development and functionality of the gastrointestinal tract, including histomorphology, immune, and endocrine systems, may be modulated by dietary components per se (ingredients, nutrients, and additives).

Poultry feed ingredients include a vast range of varying nutrients, non-nutrients, antinutrients, and beneficial and potentially harmful components. On the other hand, the chickens’ digestive tract contents mainly consisted of ingested feed ingredients, transient microbial populations, and endogenous secretions from the source of organs such as the liver, gall bladder, and pancreas. The GIT must selectively allow the nutrients while preventing the harmful components of the diet from crossing the intestinal barrier and entering the blood. In addition to simply preventing access to the bird by blocking entrance, immune tissues and cells within the gut actively respond to microbial challenges [39]. Gut microflora has been proven to significantly affect boiler nutrition, health, and growth performance [40] by interacting with nutrient utilization and GIT development in the host [41]. It seems that the use of moderate-temperature processing by eliminating harmful bacteria and removing anti-nutritive factors can be effective in maintaining intestinal microbial balance and preventing dysbiosis, and as a result, it can lead to an increase in the population of beneficial bacteria. Moreover, by improving the efficiency of the mucosal immune system, it can be effective in improving the health of the bird’s intestines, and thus improving the bird’s health.

### 4.4. Relative Length of the Small Intestine and Jejunum Morphology

The results of the current study revealed that the duodenum, jejunum and cecum lengths of birds fed the CT_U_-treated corn grains were greater than those fed the CT_75_ and CT_90_. Gut development has been shown to effectively depend on the particle size of structural components [42], where high coarseness is required in order to have a considerably positive effect on gut health [43]. Recent investigations showed that coarsely ground corn had significantly positive impacts on gut development [31]. The current study showed that the length of the digestive tract was not influenced by corn PS. However, the CT had a significant effect on the length of the duodenum, jejunum and cecum, so that the chickens that were fed with CT_U_ diets had the highest length of the duodenum, jejunum and cecum. Except for the caeca, digesta has been shown to be retained in the GIT for three to four hours [44]. So, it is assumed that the digesta possibly spends only 60 to 90 min in the anterior parts of the digestive tract, which gives only a limited opportunity for enzyme action [45]. According to the studies, as the passage rate increases, the duration time of digestion and absorption reduces in GIT, while on the other hand, a slow passage rate has been shown to limit feed intake [46]. Several factors are known to affect the passage rate, including the chicken strain [47], age [48], dietary NSP contents [49], water insoluble NSP proportion [50], dietary fat level [51], and environmental temperature [47]. In general, larger particles retain longer than finer particles in the digestive tract [22,50]. Thus, the proportion of coarse fiber in the gizzard is double that in the feed [42], possibly reflecting selective retention of coarse particles [42,52]. In fact, non-hydro-thermal conditioned diets increased the length of small intestine segments in comparison to those conditioned at 75 and 90 °C. Also, a significant PS×CT interaction was observed, where PS_F_-CT_75_ reduced and PS_C_-CT_75_ increased cecum length compared to the other experimental groups.

The poultry digestive tract development, especially the gizzard section, has been shown to be affected by feed PS, which is considerable during the first seven days [6]. In the present research, the relative weight of the gizzard, liver, proventriculus and pancreas was not significantly affected by diets containing different PS-treated corn. In the current study, the CT had no significant impact on the gizzard and liver weights. However, the chickens fed with CT_U_ diets had a higher pancreas weight compared to the other groups (CT_75_ and CT_90_). Ghobadi and Karimi [53] indicated that feeding broiler chickens with processed feeds (pelleted vs. mash) from 1 to 36 d of age significantly affected the pancreas weight. Also, an orthogonal comparison showed that birds fed diets containing hydrothermally processed corn had significantly higher relative gizzard weights compared to those fed unprocessed corn. There are conflicting reports regarding the effects of pelleted feeds and unprocessed mash diets on the gizzard weight of broiler chickens, where some of the studies showed a reduction and some showed no significant impacts of pelleting on gizzard weight [38].

The current study revealed a significant difference in muscle thickness, crypt depth, and villus width of the jejunum in broiler chickens fed PS_C_-treated corn. However, villus height (VH), VH to crypt depth (CD) ratio, VH to villus width (VW) ratio, and absorption surface area were significantly reduced by feeding coarse particle size compared to the fine one. GIT development is crucial in modern broiler production since nutrient utilization strongly depends on it. Researchers have reported that broilers fed a conditioned diet had a lower relative length of the digestive tract segments compared to those fed with a mash diet [54,55]. Amerah et al. [6] and Zang et al. [56] observed higher VH and CD in the jejunum of the birds fed a processed diet compared to those fed an unprocessed mash diet, while the earlier studies had reported a lower GIT segment length by feeding diets containing conditioned corn grain. The area for nutrient absorption increases as the VH increases in the small intestine segment. In the meantime, a higher villus absorptive area increases digestive enzyme action and nutrient transportation at the villus surface [57]. Conditioning the diet at 88 °C reduces the cecal weight by 10.7% in comparison to those conditioned at 60 °C. Higher dietary fermentable ingredients result in an enlarged caecum [58]. Svihus et al. [59] hypothesized that fermentable ingredient passage is impeded by conditioning the broiler diets at 88 °C, which in turn may result in lower relative cecal weight. Also, they declared the relative duodenum and jejunum length increased by 7.5 and 7.3%, respectively, by conditioning diets at 88 °C compared to those conditioned at 60 °C. Consistent with the previously mentioned findings, Abdollahi et al. [27] reported that conditioning broiler diets at 75 °C and 90 °C increased small intestine length by 6.3% compared to those conditioned at 60 °C. The most recent studies have considered it the natural body response to reduced nutrient availability in higher CT-exposed diets [60].

### 4.5. Gizzard pH

The gizzard plays a key role in the poultry digestive tract, and the studies showed that diet structure may effectively stimulate its development and functioning. It was the reason why diet and physical form regained intense interest in poultry nutrition. According to the close interaction between the secretory proventriculus and the functional gizzard, the efficiency with which pepsin and hydrochloric acid will degrade feed nutrients will be dependent on the functionality of the gizzard in terms of contraction intensity and retention time. Also, since the low pH caused by the hydrochloric acid is considered to potentially have a beneficial effect on gut health through its sterilizing properties, the functionality of the gizzard may also affect gut health [44]. The gizzard pH in chickens that were fed with coarsely ground corn was significantly lower than that of chickens fed with finely ground corn. This can be due to the fact that the coarse particles of corn stimulate the secretion of acid in the gizzard and lead to a lower gizzard pH. This finding is consistent with that of the other studies, where Nir et al. [22] indicated that the seven-day-old chicks fed medium or coarse particle-size diets exhibited a lower gizzard pH and higher gizzard development compared to those fed with fine particle-sized diets. The pH of gastric juice secreted by the proventriculus has been reported to be around 2. However, the amount, retention time, and chemical characteristics of the feed in the gizzard/proventriculus region will result in a more variable and usually higher pH. In a recent study, the pH of gizzard contents from broiler chickens varied between 1.9 and 4.5, with an average value of 3.5 [44]. In this study, the CT did not significantly affect gizzard pH.

## 5. Conclusions

The results of the present study confirmed that using fine PS of corn improved FCR at starter, finisher, and whole period. Also, conditioning at 75 °C improved weight gain in the starter, grower, and finisher periods, while increasing the condition temperature had no significant effect on performance compared to the unconditioned diet. Based on the current findings, using a fine particle size of corn conditioned at 75 °C can improve the growth performance and gastrointestinal tract development of broiler chickens. Accordingly, further studies should be conducted by researchers related to this subject to specify the effect of different PS on different rearing phases (starter, grower and finisher).

## Figures and Tables

**Table 1 animals-14-00818-t001:** Ingredients and calculated chemical composition (%) of the experimental basal diet.

Ingredients	Starter (0–10 Days)	Grower (11–24 Days)	Finisher (25–42 Days)
Corn (7.80% CP)	48.90	52.64	57.74
Soybean meal (43.88% CP)	42.75	38.74	33.31
Soybean oil	4.50	5.20	5.81
Dicalcium phosphate	1.13	0.92	0.75
Calcium carbonate	1.07	0.99	0.92
Common salt	0.33	0.33	0.33
Sodium bicarbonate	0.15	0.13	0.13
Mineral premix ^a^	0.25	0.25	0.25
Vitamin premix ^a^	0.25	0.25	0.25
DL-Methionine	0.33	0.28	0.26
L- Lysine HCL	0.21	0.16	0.16
L- Threonine	0.07	0.05	0.03
Choline Chloride	0.05	0.05	0.05
Phytase (1000 FTU/kg)	0.01	0.01	0.01
Calculated chemical composition			
Apparent metabolizable energy (kcal/kg)	3000	3100	3200
Crude protein (%)	23.00	21.50	19.50
Calcium (%)	0.96	0.87	0.79
Available phosphorus (%)	0.48	0.435	0.395
Potassium (%)	0.98	0.92	0.83
Chlorine (%)	0.25	0.25	0.24
Sodium (%)	0.18	0.18	0.18
Digestible Lysine (%)	1.28	1.15	1.03
Digestible Methionine (%)	0.63	0.57	0.53
Digestible Methionine + Cystine (%)	0.95	0.87	0.80
Digestible Threonine (%)	0.86	0.77	0.72
Digestible Tryptophan (%)	0.25	0.24	0.21
Digestible Arginine (%)	1.52	1.42	1.26
Digestible Isoleucine (%)	0.92	0.86	0.77
Digestible Leucine (%)	1.77	1.68	1.55
Digestible Valine (%)	0.98	0.93	0.84

^a^ Supplied per kilogram of diet: antioxidant, 100 mg; biotin, 0.2 mg; calcium pantothenate, 20 mg; cholecalciferol, 4500 IU; cyanocobalamin, 0.017 mg; folic acid, 2.0 mg; menadione, 3.6 mg; niacin, 65 mg; pyridoxine, 4 mg; trans-retinol, 12000 IU; riboflavin, 8 mg; thiamine, 4.0 mg; all-rac-α-tocopheryl acetate, 80 IU; Cu, 16 mg; Fe, 20 mg; I, 1.25 mg; Mn, 120 mg; Se, 0.3 mg; Zn, 120 mg.

**Table 2 animals-14-00818-t002:** Effect of corn particle size (PS) and conditioning temperature (CT) on the growth performance of broiler chickens.

Treatments		Starter (1–10 Days)			Grower (11–24 Days)			Finisher (25–42 Days)			Total (1–42 Days)		
PS *	CT (°C)	ADFI ^1^ (g/Bird/d)	ADWG ^2^ (g/Bird/d)	FCR ^3^ (g/g)	ADFI (g/Bird/d)	ADWG (g/Bird/d)	FCR (g/g)	ADFI (g/Bird/d)	ADWG (g/Bird/d)	FCR (g/g)	ADFI (g/Bird/d)	ADWG (g/Bird/d)	FCR (g/g)
F	-	30.9	22.7 ^a^	1.360 ^b^	73.7	47.6	1.552	142	89.5	1.592 ^b^	89.1	57.3 ^a^	1.554 ^b^
C	-	29.7	20.8 ^b^	1.433 ^a^	74.8	47.4	1.582	143	87.8	1.638 ^a^	89.8	56.1 ^b^	1.600 ^a^
SEM	-	0.404	0.358	0.0187	0.642	0.564	0.0205	0.850	0.823	0.0137	0.495	0.395	0.0116
-	U	29.5	20.6 ^b^	1.439 ^a^	73.4	45.0 ^b^	1.634 ^a^	142	89.0	1.602 ^ab^	88.7	55.7 ^b^	1.594
-	75	30.4	22.4 ^a^	1.354 ^b^	75.0	48.2 ^a^	1.555 ^b^	142	89.7	1.593 ^b^	89.7	57.6 ^a^	1.557
-	90	31.0	22.2 ^a^	1.396 ^ab^	74.5	49.3 ^a^	1.512 ^b^	143	87.2	1.649 ^a^	89.9	56.9 ^ab^	1.580
SEM		0.494	0.439	0.0229	0.787	0.691	0.0252	1.041	1.009	0.0168	0.607	0.484	0.0142
F	U	30.4	21.5	1.414	73.3	44.8	1.640	141	90.4	1.567	88.6	56.4	1.571
	75	31.2	23.5	1.325	73.9	48.3	1.528	142	91.2	1.562	89.3	58.4	1.527
	90	31.0	23.1	1.341	74.0	49.7	1.488	143	86.9	1.647	89.4	57.1	1.564
C	U	28.7	19.6	1.464	73.4	45.2	1.628	143	87.5	1.637	88.8	54.9	1.618
	75	29.6	21.4	1.383	76.1	48.1	1.581	143	88.3	1.624	90.1	56.7	1.588
	90	30.9	21.4	1.452	75.0	48.8	1.536	144	87.6	1.652	90.4	56.6	1.595
SEM		0.699	0.621	0.0323	1.113	0.977	0.0356	1.472	1.427	0.0237	0.858	0.685	0.0200
*p* Value													
PS		0.064	<0.001	0.010	0.236	0.758	0.341	0.250	0.152	0.026	0.321	0.039	0.009
CT (°C)		0.146	0.010	0.048	0.351	<0.001	0.007	0.617	0.226	0.057	0.362	0.031	0.204
PS × CT (°C)		0.477	0.973	0.601	0.634	0.809	0.604	0.969	0.382	0.343	0.901	0.641	0.776

* Abbreviations: PS, particle size; CT, conditioning temperature; ^1^ ADFI, Average Daily Feed Intake; ^2^ ADWG, Average Daily Weight Gain; ^3^ FCR, Feed Conversion Ratio; U, unconditioned; 75, conditioned at 75 °C; 90, conditioned at 90 °C; F, Fine particle size of corn grain; C, Coarse particle size of corn grain. a–b Means within a column with different superscripts differ significantly (*p* < 0.05).

**Table 3 animals-14-00818-t003:** Effect of corn particle size and conditioning temperature on carcass traits, gastrointestinal organ weights and gizzard pH of broiler at 42 days.

PS *	CT (°C)	Slaughtered Body Weight (g)	Carcass Yield	Breast	Thigh	Abdominal Fat	Gizzard	Liver	Pancreas	Heart	Gizzard pH
			% of Live Weight								
F	-	2522 ^a^	67.8	26.2	20.1	2.036 ^b^	1.033	1.906	0.2215	0.459	3.978
C	-	2472 ^b^	67.5	25.7	20.3	2.412 ^a^	1.171	1.852	0.2002	0.479	3.125
SEM	-	16.8	0.28	0.34	0.27	0.1065	0.0219	0.0568	0.0047	0.0167	0.0640
-	U	2455	67.6	25.7	20.4	2.142	1.230	1.949	0.2318	0.479	3.623
-	75	2524	67.9	26.2	20.1	2.180	1.061	1.783	0.2072	0.454	3.513
-	90	2512	67.4	25.9	20.0	2.320	1.016	1.904	0.1936	0.475	3.517
-	SEM	20.6	0.34	0.42	0.33	0.1304	0.0268	0.0696	0.0058	0.0205	0.0784
F	U	2492	67.3 ^b^	26.0	20.0	1.950	1.132 ^bc^	2.170 ^a^	0.2378 ^a^	0.454	4.060 ^a^
	75	2562	68.9 ^a^	26.3	20.6	1.975	1.071 ^bcd^	1.813 ^ab^	0.2415 ^a^	0.457	3.740 ^ab^
	90	2512	67.1 ^b^	26.3	19.6	2.184	0.897 ^d^	1.733 ^ab^	0.1853 ^b^	0.467	4.135 ^a^
C	U	2418	67.9 ^ab^	25.5	20.8	2.335	1.328 ^a^	1.728 ^b^	0.2258 ^ab^	0.504	3.187 ^bc^
	75	2485	67.7 ^ab^	26.1	19.6	2.455	1.051 ^cd^	1.832 ^ab^	0.1728 ^b^	0.451	3.287 ^bc^
	90	2513	66.8 ^b^	25.5	20.5	2.456	1.135 ^bc^	1.995 ^ab^	0.2019 ^ab^	0.483	2.900 ^c^
SEM		29.2	0.49	0.60	0.47	0.1844	0.0379	0.0984	0.0082	0.0289	0.1109
*p* Value											
PS		0.045	0.471	0.371	0.568	0.042	<0.001	0.554	0.010	0.455	<0.001
CT		0.060	0.705	0.780	0.731	0.692	<0.001	0.317	0.002	0.718	0.610
PS × CT		0.337	0.034	0.927	0.190	0.889	0.014	0.022	<0.001	0.697	0.019

* Abbreviations: PS, particle size; CT, conditioning temperature; F = Fine particle size of corn grain, C = Coarse particle size of corn grain, U = Unconditioned, 75 = Conditioned at 75 °C and 90= Conditioned at 90 °C. a–d Means within a column with different superscripts differ significantly (*p* < 0.05).

**Table 4 animals-14-00818-t004:** Effect of corn particle size and conditioning temperature on SRBC and Newcastle disease (ND) titer of broiler.

PS *	CT (°C)	Primary Responses			Secondary Responses			ND Titer
		Ig T	IgG	IgM	Ig T	IgG	IgM	
F	-	4.533	1.866	2.666	5.666	2.733	2.933	6.666
C	-	5.066	1.933	3.133	6.333	3.066	3.266	7.066
SEM	-	0.3000	0.1944	0.2963	0.3073	0.3448	0.2186	0.3055
-	U	4.300 ^b^	1.700	2.600 ^b^	5.200 ^b^	2.600	2.600 ^b^	6.800
-	75	5.500 ^a^	1.900	3.600 ^a^	6.600 ^a^	2.800	3.800 ^a^	7.100
-	90	4.600 ^ab^	2.100	2.500 ^b^	6.200 ^ab^	3.300	2.900 ^b^	6.700
-	SEM	0.3674	0.2380	0.3629	0.3764	0.4223	0.2677	0.3742
F	U	4.000	1.600	2.400	5.000	2.400	2.600	6.400
	75	5.400	2.000	3.400	6.200	3.000	3.200	6.600
	90	4.200	2.000	2.200	5.800	2.800	3.000	7.000
C	U	4.600	1.800	2.800	5.400	2.800	2.600	7.200
	75	5.600	1.800	3.800	7.000	2.600	4.400	7.600
	90	5.000	2.200	2.800	6.600	3.800	2.800	6.400
SEM		0.5196	0.3367	0.5132	0.5323	0.5972	0.3786	0.5292
*p* Value								
PS		0.220	0.810	0.276	0.138	0.500	0.291	0.363
CT(°C)		<0.01	0.503	<0.01	<0.01	0.492	<0.01	0.736
PS × CT(°C)		0.842	0.792	0.975	0.910	0.510	0.157	0.276

* Abbreviations: PS, particle size; CT, conditioning temperature; F = Fine particle size of corn grain, C = Coarse particle size of corn grain, U = Unconditioned, 75 = Conditioned at 75 °C and 90 = Conditioned at 90 °C; ND, Newcastle disease. a–b Means within a column with different superscripts differ significantly (*p* < 0.05).

**Table 5 animals-14-00818-t005:** Effects of corn particle size and conditioning temperature on relative length (cm/kg body weight) jejunum morphology of the digestive tract of broiler at 42 days.

PS *	CT (°C)	Duodenum	Jejunum	Ileum	Total Small Intestine	Cecum	Villus Height (µm)	Villus Width (µm)	Villus Height/Crypt Depth	Crypt Depth (µm)	Muscle Thickness (µm)	Absorption Surface Area (µm)
		(cm/kg Body Weight)										
F	-	12.6	30.9	31.0	75.3	7.89	1080	100.4 ^b^	12.77	87.8	134	344,090
C	-	13.3	29.4	31.0	73.1	7.93	833	117.4 ^a^	8.55	105.1	150	306,872
SEM	-	0.392	0.546	0.810	1.500	0.1550	34.0	4.95	0.710	5.34	6.0	17,340
-	U	14.2 ^a^	32.7 ^a^	31.4	78.4 ^a^	8.46 ^a^	918	99.4 ^b^	10.10	98.2	166	284,167
-	75	12.0 ^b^	27.8 ^b^	30.2	70.1 ^b^	7.40 ^b^	986	113.5 ^a^	10.18	106.3	133	348,582
-	90	12.7 ^ab^	29.9 ^ab^	31.3	74.0 ^ab^	7.87 ^ab^	965	113.8 ^a^	11.71	84.8	128	343,695
-	SEM	0.480	0.668	0.992	1.837	0.1899	41.6	6.06	0.870	6.54	7.3	21,237
F	U	13.9	32.6	30.0	76.6	8.67 ^a^	937 ^b^	88.3	12.11 ^a^	79.1 ^b^	116 ^c^	257,873 ^c^
	75	11.9	25.7	30.0	67.7	6.91 ^b^	1244 ^a^	110.0	14.05 ^a^	94.5 ^ab^	156 ^b^	430,018 ^a^
	90	12.0	30.1	32.8	75.0	8.11 ^ab^	1057 ^ab^	102.8	12.15 ^a^	89.8 ^ab^	129 ^bc^	344,379 ^b^
C	U	14.5	32.9	32.7	80.2	8.25 ^ab^	899 ^ab^	110.6	8.09 ^b^	117.4 ^a^	215 ^a^	310,459 ^bc^
	75	12.0	30.0	30.4	72.5	7.90 ^ab^	728 ^c^	116.9	6.31 ^b^	118.0 ^a^	109 ^c^	267,145 ^bc^
	90	13.4	29.8	29.8	73.1	7.64 ^ab^	872 ^bc^	124.7	11.27 ^a^	79.9 ^b^	127 ^bc^	343,009 ^b^
SEM		0.680	0.945	1.403	2.598	0.2685	58.9	8.57	1.231	9.26	10.4	30,034
*p* Value												
PS		0.279	0.099	0.995	0.366	0.891	<0.001	0.001	<0.001	0.002	0.008	0.036
CT (°C)		0.027	0.0008	0.697	0.034	<0.01	0.263	0.032	0.124	0.006	<0.001	0.005
PS × CT (°C)		0.68	0.086	0.208	0.481	0.042	<0.001	0.357	0.001	0.001	<0.001	<0.001

* Abbreviations: PS, particle size; CT, conditioning temperature; F = Fine particle size of corn grain, C = Coarse particle size of corn grain, U = Unconditioned, 75 = Conditioned at 75 °C and 90 = Conditioned at 90 °C. a–c Means within a column with different superscripts differ significantly (*p* < 0.05).

## Data Availability

The data that support the findings of this study are available from the corresponding author, Asadollah Ghasemi-Aghgonbad, upon reasonable request.

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
