# Peer review of "The Interactive Impacts of Corn Particle Size and Conditioning Temperature on Performance, Carcass Traits, and Intestinal Morphology of Broiler Chickens"

_animals, 2024, doi:10.3390/ani14050818_

Round 1

Reviewer 1 Report

Comments and Suggestions for Authors

It is generally well-written manuscript, I have only some minor comments. Please see the attachment.

Author Response

Thanks for your valuable comments on my paper and thank you in advance for your consideration. We corrected most parts of the manuscript according your suggestions. However, I have explanations for some parts of the manuscript. Hope it will be acceptable for you. We have added simple summary and references are formatted according to the journal's specifications. All abbreviations explained.  In all tables we have added three decimal places and we use "g" instead of "gr" for gram. The conclusion section has been rewritten.

Reviewer 2 Report

Comments and Suggestions for Authors

The article is well written and structured. I suggest some changes:

Authors must adapt the manuscript to the journal's standards. References must be made in the text in vancouver style.

The results in the abstract should be better presented. Just mentioning that a significant effect occurred leaves many openings, so the treatments with the greatest or least results must be expressed.

A recommendation at the conclusion of the abstract is necessary

Remove from the keywords the words that were mentioned in the title.

Material and methods:

The research ethics committee protocol number with the number of animals must be entered

Enter the equations used to calculate ADWG (Average Daily Weight Gain), ADFI (Average Daily Feed Intake), and FCR More details about the slaughter of the animals are needed.

The authors must detail all the steps and procedures adopted. Did you follow any methodology? Were the animals fasting? Were animal welfare standards followed?

Standardize the equations in the text, observe the journal template

The statistical model must be inserted

The conclusion needs to be rewritten. The form presented is a continuation of the discussion of the data. The conclusion must be a response to the research objective and hypothesis

Author Response

References were formatted according to the journal's specifications (Vancouver style). Keywords have been rewritten. The research ethics committee protocol number and date added in beginning of material and method. The conclusion section (was add a recommendation) and statistical model have been rewritten. The equations were used to calculate ADWG, ADFI, and FCR have added. The equations in the text were Standardized following the journal template. In all the steps and procedures of the research, according to the ethical approval certificate of the university and the scientific and animal welfare principles, all the basic principles, including breeding density, lightening, and 6 hours of starvation before slaughter, were applied. To allow for emptying the gastrointestinal tract contents, the chickens were slaughtered after six hours of fasting, but water was made available to the chickens.

The authors confirm that they have followed EU standards for the protection of animals used for scientific purposes and feed legislation. All experimental procedures of care and use of animals were approved by the Research Ethics Committees of University of Tabriz, Iran (Approval ID:  IR.TABRIZU.REC.1402.130).

Reviewer 3 Report

Comments and Suggestions for Authors

The present study investigated the interactive impacts of corn particle size and conditioning temperature on performance, carcass traits and interstinal morphology of broilers.  The present study may benefit readers focusing on poultry production, but there are some limitations. 

Major concerns:

Indeed, this study was not a new topic. In the introduction, more backgroud about the fingdings in previous studies about the effects of corn particle size or conditioning temperatures on poultry should be given. 

Nutrients digestion rates were not determined in this study.   It is clear that corn particle size and contionting temperature will affect nutrients digestion rates and result in difference in growth performance. 

Also, the activiety of digestive enzymes like protease, lipase and so on in chyme should also be determined. 

The authors stated in the MM that they measured the villus height, crypt depth, and Villus absorption surface of jejunum. But these results were not provided. 

Author Response

We have added simple summary and references were formatted according to the journal's specifications. In all tables we have added three decimal places. we were used "g" instead of "gr" for gram. In order to measure the muscle layer thickness of broiler intestine, the circular and longitudinal muscular layers thickness was measured at 10- times objective by using a light micro-scope (Olympus BX 41, Japan). The conclusion section and statistical model have been rewritten. The activity of digestive enzymes were determined and will report in future Article about nutrient digestibility. The villus height, crypt depth, and Villus absorption surface of jejunum data was showed in table5 at page number of 8.

Reviewer 4 Report

Comments and Suggestions for Authors

This experiment by Ghasemi-Aghgonbad et al., investigate the interactions between corn particle size (PS) and conditioning temperature (CT) on  the growth performance, carcass traits, intestinal morphology, and immune responses in broilers fed a corn-soybean based diet. Overall, the topic selection has guiding significance for poultry production, with reasonable experimental design and feasible research methods. I have only some comments as follows.

The title including intestinal morphology, but there were no the related results in the manuscript. It just give the results of the relative length of the digestive tract.

There need provide the result of jejunum morphology, becasue there have the description of this results in Results and Discussion section.

Introduction and the Methods and Merterial: When first apperance of the abbreviation, it need give the full name, such as FCR, CT, PS.

The title of Table 2. change "performance" to "growth performance".

The title of Table 5. deleted the  jejunum morphology.

In the tables. What the "F", "C"and "U" means?

All of the "P" need "P".

Author Response

The result of jejunum morphology was provided in table 5 at page 8. the full name of all abbreviation were mentioned at the first appearance place. The title of Table 2 was changed ("performance" to "growth performance"). All the "P" was changed to "P".

Round 2

Reviewer 3 Report

Comments and Suggestions for Authors

The manuscript has been improved and could be considered for publication.